# Changes in Growth, Ionic Status, Metabolites Content and Antioxidant Activity of Two Ferns Exposed to Shade, Full Sunlight, and Salinity

**DOI:** 10.3390/ijms24010296

**Published:** 2022-12-24

**Authors:** Anna Pietrak, Piotr Salachna, Łukasz Łopusiewicz

**Affiliations:** 1Department of Horticulture, West Pomeranian University of Technology in Szczecin, 71-459 Szczecin, Poland; 2Center of Bioimmobilisation and Innovative Packaging Materials, West Pomeranian University of Technology in Szczecin, 71-270 Szczecin, Poland

**Keywords:** environmental stress, peridophytes, physiological responses, antioxidant activity

## Abstract

The interactions between ferns and the environment have been frequently researched. However, detailed data on how ferns respond to specific stresses and a combination of stress factors during cultivation are lacking. This study assessed the effects of salinity and full sunlight and the combination of both stresses on the growth and selected metabolic parameters of two hardy ferns (*Athyrium nipponicum* cv. Red Beauty and *Dryopteris erythrosora*) under production conditions. Hardy ferns are highly interesting ornamental plants that can serve as a potential source of antioxidants for the pharmaceutical, cosmetic, and food industries. The results showed that in both ferns, salinity and salinity combined with full sunlight lowered the dry weight of the aerial part and potassium/sodium and calcium/potassium ratio compared with control plants. Salinity, full sunlight, and multi-stress did not affect the total polyphenol content in both ferns but increased the total free amino acids and flavonoids in *D. erythrosora*. In *A. nipponicum* cv. Red Beauty, all stressors decreased the total free amino acids content and the antioxidant activities determined by ABTS, DPPH, FRAP, and reducing power assays. By contrast, plants of *D. erythrosora* grown under full sunlight are characterized by higher antioxidant activities determined by DPPH, FRAP, and reducing power assays. Overall, a greater adaptive potential to abiotic stresses was found in *D. erythrosora* than in *A. nipponicum* cv. Red Beauty. Our findings shed some light on the physiological mechanisms responsible for sensitivity/tolerance to salinity, full sunlight, and combined stresses in hardy ferns.

## 1. Introduction

The first species of ferns appeared about 360 million years ago at the end of the Paleozoic era. The presently known at least 10,000 fern species are probably only a tiny fraction of the very rich group of plants common at the time [1,2]. During their long evolution, ferns have adapted to various, highly diverse habitats. They can be found in almost all warm and cold climate zones. They grow in shady rainforests in equatorial and subtropical regions, along the banks of streams and rivers, and in sunny locations in the mountains, open meadows, and dry and desert areas [3]. Ferns have great economic importance. They are grown as ornamental and edible plants, are used in pharmacology and phytomedicine, and are effective in land reclamation [4,5]. Hardy ferns are becoming increasingly popular ornamental plants, including several species and varieties recommended for home gardens, rock gardens, ponds, and parks. Hardy ferns can be deciduous, winter-green, semi-evergreen, or evergreen, which makes them highly sought-after ornamental plants [6]. Some species of hardy ferns are rich sources of antioxidants with potential use in medicine, pharmacy, cosmetics, and the food industry [5,7]. Most ferns require moist and shady habitats, but there are also taxa tolerant of dry and sunny locations [6,8]. So far, no specific technologies have been developed for producing hardy ferns grown for both ornamental and medicinal purposes. There are also no detailed data on how ferns respond to specific stresses and a combination of stress factors during cultivation.

Plants are constantly exposed to abiotic stresses caused by chemical and physical factors. An example of a chemical factor evoking plant stress is excessive soil salinity [9]. Light stress is a type of stress caused by a physical factor in the form of excess light [10,11]. The consequence of salt and light stress is reduced plant growth, productivity, quality, and oxidative damage due to the overproduction of reactive oxygen species (ROS) [12]. The cellular level of ROS is controlled by a complex network of antioxidants that includes substances alleviating the effects of oxidative stress, such as low molecular weight antioxidants, polyphenols, flavonoids, carotenoids, and some free amino acids [13].

There are numerous studies on plant responses to single stress factors. However, the effects of combined stresses on plant physiological and metabolic processes have been much less studied [14,15]. The mechanisms of resistance to environmental stresses are known mainly in seed plants, and not much is known about how non-flowering plants, including hardy ferns, respond to abiotic stresses. *Athyrium nipponicum* (Mett.) Hance (family *Athyriaceae*) is a representative of deciduous ferns. Due to the colorful leaf innervation, the cultivars of this species are considered the most ornamental of all hardy ferns. *A. nipponicum* leaves contain sulfolipids with anti-HIV potential [16]. An example of hardy evergreen ferns is *Dryopteris erythrosora* (DC Eaton) Kuntze (family Dryopteridaceae). The species is distinguished by its orange leaves, rarely found in other ferns. Apart from its decorative value, *D. erythrosora* is a valuable source of flavonoids showing specific biological activity, including antioxidant and antineoplastic potential [17,18]. This study assessed the effects of salinity, full sunlight, and a combination of these stresses on the growth, mineral status, the content of selected primary and secondary metabolites, and antioxidant activity of *A. nipponicum* and *D. erythrosora*. We assumed that adverse environmental factors would affect plant growth, the content of metabolites, and the antioxidant potential of hardy ferns.

## 2. Results 

### 2.1. Effect of Abiotic Stresses on Fern Growth Parameters

The studies showed changes in plant morphology depending on the treatment and fern taxon (Table 1). Plants of *A. nipponicum* cv. Red Beauty exposed to salinity stress in shade conditions were lower (by 19%) and had a reduced fresh weight of the aboveground part (by 16.3%) compared to control plants. Moreover, the salted plants had shorter fronds (by 18.9%). The exposure of plants to full sunlight (cultivation without shade) did not affect plant height or biomass. Under conditions of salinity and full sunlight, *A. nipponicum* cv. Red Beauty was lower (by 27.3%) and had a reduced fresh weight of the aboveground part (by 16.3%). In the case of *D. erythrosora*, salinity in shade conditions significantly reduced plant height (by 37.6%), frond length (by 27.7%), and fresh weight of the aboveground part (by 41.7%) compared to the control. When the plants were grown without shade in full sun, they were found to be taller (by 19.6%) and had increased biomass (by 36.4%) compared to untreated plants. Under salinity combined with no shading conditions, *D. erythrosora* plants had similar height and frond length, while the fresh weight of the aboveground part was reduced (by 31.8%) compared to the control. 

In the case of *D. erythrosora*, no browning or drying of the leaf blades was observed in either fern taxa under the influence of salinity (Appendix A). Furthermore, the exposure of the plants to direct sunlight did not cause damage or discoloration of the leaf blades of the plants. Little visual leaf damage under conditions of salt stress was observed in *A. nipponicum* cv. Red Beauty (Appendix A).

### 2.2. Effect of Abiotic Stresses on Fern Ion Balance, Chlorophyll Content, Carbohydrates, and Amino Acid Metabolism

The values of the K^+^/Na^+^ and Ca^2+^/Na^+^ coefficients decreased drastically in both fern taxa due to the applied stress factors (Table 2). The most significant decrease in the K^+^/Na^+^ and Ca^2+^/Na^+^ coefficients was noted in both ferns subjected to salinity stress in the shade and full sunlight conditions. In plants cultivated without screens, reduced K^+^/Na^+^ and Ca^2+^/Na^+^ coefficients were also noted compared with the control, with statistically confirmed differences for the K^+^/Na^+^ coefficient in *A. nipponicum* and the Ca^2+^/Na^+^ coefficient in *D. erythrosora*.

Because of stress factors, there were significant changes in the concentration of assimilation pigments depending on the fern genotype (Table 3). In *A. nipponicum* cv. Red Beauty, exposure of plants to salinity under shade conditions increased chlorophyll a, chlorophyll b, and total chlorophyll by 14.3%, 21.1%, and 17.0%, respectively, over the control. Alternatively, plants growing in full sunlight significantly decreased chlorophyll a, chlorophyll b, and total chlorophyll by 27.8%, 22.5%, and 26.0%, respectively, compared to the control. Similarly, decreases in the content of chlorophyll a, chlorophyll b, and total chlorophyll by 39.9%, 33.8%, and 37.9%, respectively, were shown in the salted plants growing without screens. In the case of *D. erythrosora*, chlorophyll a, chlorophyll b, and total chlorophyll decreased by 27.8%, 22.5%, and 26.0%, respectively, concerning unstressed plants. Salinity in the shade and full sunlight conditions did not affect the content of pigments in the leaves of *D. erythrosora*. The analysis of chlorophyll composition revealed significant differences in the Chl a/b ratio depending on the exposure of plants to stress. In *A. nipponicum* cv. Red Beauty, the Chl a/b ratio was lower than the control in all variants using stress factors. In *D. erythrosora*, salinity in shade conditions did not affect the value of the Chl a/b ratio, whereas the exposure of plants to full sunlight and total sun exposure combined with salinity resulted in a significant decrease in the Chl a/b ratio.

Significant changes were found in the content of reducing sugars and total free amino acids in fern leaves depending on the applied stress factors (Figure 1a,b). In *A. nipponicum* cv. Red Beauty, salinity stress in the shade and full sunlight conditions increased the synthesis of reducing sugars compared to the control. In turn, a decrease in the content of reducing sugars was shown in plants growing in full sunlight. In *D. erythrosora*, the amount of reducing sugars was significantly reduced under all applied stress variants, with the most significant decrease observed in plants exposed to salinity under full sunlight conditions. Regardless of the treatment, the leaves of *D. erythrosora* contained more reducing sugars than the leaves of *A. nipponicum* cv. Red Beauty.

In *A. nipponicum* cv. Red Beauty, the content of total free amino acids in the leaves, decreased due to the application of stresses and was the lowest in salted plants growing in full sunlight conditions. The opposite trend was shown in *D. erythrosora* under the influence of the tested stresses, for which the content of total free amino acids increased significantly, on average, by more than 2.7 times compared to the control (Figure 1b). 

### 2.3. Effect of Abiotic Stresses on Antioxidant Compounds and Antioxidant Properties of Ferns

Figure 2a,b shows the influence of stress factors on the content of secondary metabolites in fern leaves. In both taxa, salinity, excess light, and a combination of these factors did not cause changes in the total polyphenol content but did affect the total flavonoid level. *A. nipponicum* cv. Red Beauty showed a high content of total flavonoids in salted plants growing in the shade, whereas a reduced content of these metabolites was found in salted plants growing in full sunlight. In the case of *D. erythrosora*, all stresses resulted in a significant increase in the total content of flavonoids, with the highest concentration of metabolites in plants exposed to full sunlight. Comparing both taxa, extracts from *D. erythrosora* leaves contained more total polyphenols and flavonoids than extracts from *A. nipponicum* cv. Red Beauty.

Analyses of the antioxidant potential of fern extracts grown under stress conditions are shown in Table 4. In *A. nipponicum* cv. Red Beauty, salinity, full sunlight, and the combined use of these stress factors resulted in a significant reduction in antioxidant activity measured by the ABTS, DPPH, FRAP, and reducing power tests. Deal stress impacted the neutralization of free radicals determined by the ABTS method. Multistress caused the most significant drop in ABTS, followed by full sunlight and saline. In *D. erythrosora*, the values of the ABTS radical were not affected by stress factors, whereas in the case of the second assessed radical, DPPH, significant differences were already demonstrated. Complete sunlight treatment gave the highest values for DPPH, followed by control, saline, and multi-stress. Similarly, the highest FRAP values were obtained in plants under full sunlight, followed by control, multistress, and saline. Moreover, extracts from plants exposed to full sunlight had the highest antioxidant activity measured by reducing power. The two investigated ferns demonstrated a significant correlation (Appendix A) between the total free amino acids content and the results of reducing power (r = 0.84 for *A*. *nipponicum* cv. Red Beauty and r = 0.69 for *D. erythrosora*). Moreover, in *A*. *nipponicum* cv. Red Beauty, the content of free amino acids also correlated with total phenolic content (r = 0.71) as well as the results of antioxidant activity determined by the ABTS (r = 0.75) and DPPH (r = 0.86) assays. In *A*. *nipponicum* cv. Red Beauty, equally, a significant correlation was found for total flavonoids and the results of ABTS (r = 0.80) and total polyphenols and the results of DPPH (r = 0.60).

## 3. Discussion

Plants, including ferns, are constantly exposed both in natural conditions and during cultivation to various stresses influencing changes at the morphological, physiological, and biochemical levels. Most studies on plant response to stress focus on assessing a single stress factor under strictly controlled conditions using model species [19]. Such studies may be of limited application value as many stress factors of variable intensity are usually imposed during plant production [20]. The reaction of plants to salinity in a phytotron differs from the natural variability of factors under cover or in the natural environment [21]. In these studies conducted in a polytunnel, salinity and various light conditions affected growth, ionic balance, and the content of pigments, metabolites, and antioxidant activity of two taxa of ornamental ferns rich in phytochemicals was assessed.

### 3.1. Morphology

Salinity caused by watering plants with 100 mM NaCl solution under shade conditions negatively modified the growth of both *A. nipponicum* cv. Red Beauty and *D. erythrosora* manifested by reduced plant height, shortened frond length, and reduced fresh leaf weight. The adverse effect of salt stress on the height of plants and the weight of the aboveground part is confirmed by previous studies on four ornamental species of ferns of the genus *Dryopteris* [8]. Plant growth limitation due to salinity could be related to disturbances in nutrient uptake, osmotic stress, and ion toxicity [22]. Furthermore, the high accumulation of Na^+^ ions in the tissues of glycophytes leads to reduced water absorption and, as a result, reduced plant growth [23]. Salt stress did not drastically reduce the aesthetic value of *D. erythrosora*, as no browning and drying of leaves were observed (Appendix A). No leaf chlorosis, despite salt stress, was found in ferns such as *D. filix-mas* and *D*. *filix-mas* cv. Linearis-Polydactylon [8]. It is crucial in the case of ornamental plants because the knowledge of taxa retaining decorative values despite salinity is of great practical importance [24]. There is an intensive search for ornamental plants for green areas that tolerate salinity [23], as well as species that can be watered with salty seawater without damaging their decorativeness [25]. 

Plants of *A. nipponicum* cv. Red Beauty grown in full sunlight did not differ in terms of morphological features from plants growing in the shade. A different reaction was observed in *D. erythrosora*, where plants were taller and produced more biomass under full sunlight. Both fern species under assessment grow under forests in the natural state, with *A. nipponicum* in the shade and *D. erythrosora* in half-shade conditions. The obtained results are interesting because, in both fern taxa, no symptoms of burns, whitening of leaves, or curling of leaf edges were observed when they were exposed to full sunlight (Appendix A). It should be emphasized that the research was carried out during the period when, in the climatic conditions of Poland, there is the highest insolation and the most significant number of sunny hours per year [26]. The morphological response of *D. erythrosora* to full sunlight leading to growth stimulation may result from a unique adaptive mechanism to high light intensity. Moreover, under saline conditions, *D. erythrosora* plants were taller and had longer fronds when grown without screens. However, further research is needed on which plants would be exposed to direct sunlight for a long time. Only then can it be determined whether *D. erythrosora* can be considered an ornamental plant that tolerates direct sunlight.

### 3.2. Physiological Responses

In both fern taxa, treatment with NaCl solution resulted in a significant increase in the content of Na^+^ ions and a decrease in K^+^ and Ca^2+^ ions in leaf tissues, which decreased the K^+^/Na^+^ and Ca^2+^/Na^+^ ratios. These changes have been demonstrated in both shaded and full-light plants. A high concentration of Na^+^ in leaves due to salinity, regardless of light conditions, was also demonstrated in ferns *D. atrata*, *D. affinis*, *D. filix-mas*, and *D. filix-mas* ‘Linearis-Polydactylon’ [8]. The reduced concentration of K^+^ and Ca^2+^ ions in salted plants, shown in the studies, is most likely related to the excessive accumulation of Na^+^ ions causing a cytotoxic effect [23,27]. An increase in the level of harmful Na^+^ ions in the plant increases osmotic pressure and can induce changes in the uptake of other ions, especially K^+^ and Ca^2+^, and their transport to the aboveground part. K^+^ and Ca^2+^ ions determine the activity of many enzymes and the processes of plant adaptation to stress [28]. The competition between ions leading to a decrease in K^+^/Na^+^ and Ca^2+^/Na^+^ ratios, as well as a deficit of K^+^ and Ca^2+^ in the cytoplasm, may result in the dysfunction of biological membranes and impairment of many metabolic pathways [20,23,29].

A pronounced decrease in chlorophyll content in leaf tissues distinguished plants of both taxa grown in full sunlight. As previously stated, both species prefer shaded habitats under the tree canopy, with *A. nipponicum* being more shade-loving than *D. erythrosora*. In very high light-intensity conditions, photosynthetic dyes absorb too much PAR energy, concerning the possibility of its conversion into chemical energy during photosynthesis [30]. It leads to photoinhibition and may cause oxidative stress, destroying photosynthetic pigments [31]. 

Analyzing the obtained results related to the effect of salinity on the concentration of assimilation pigments, it was shown that this stress factor stimulated chlorophyll biosynthesis in *A. nipponicum*. It has been confirmed by previous research that salinity in the initial stage and at low levels can lead to increased chlorophyll concentration in the leaves in some ferns, but at higher levels, it results in a drastic decrease in the content of dyes [32]. In the case of *D. erythrosora*, there were no changes in the concentration of assimilation dyes due to salinity. Moreover, in *D. erythrosora*, salinity did not decrease the Chl a/b ratio. Assessing the chlorophyll level in this species may be a valuable tool for assessing its adaptation to salt stress.

In *A. nipponicum* cv. Red Beauty plants, because of salinity and cultivation in full sun, the Chl a/b ratio decreased. Appropriate quantitative ratios between chlorophylls a and b determine the appropriate intensity of photosynthesis. A decrease in the Chl a/b ratio was also demonstrated in ferns *Asplenium viridae*, *Ceterach officinarum,* and *Phyllitis scolopendrium* due to exposure to high doses of NaCl (250–500 mM) [32]. The decrease in the Chl a/b ratio may be caused by a decrease in chlorophyll a’s content, which proves considerable damage to the photosystems and photoinhibition mentioned above [33,34].

Under salinity conditions, the amount of reducing sugars increased in *A. nipponicum* cv. Red Beauty, and in *D. erythrosora*, it decreased. Perhaps this relates to the different mechanisms of sensitivity/tolerance to stress in the taxa. Environmental stresses are the leading causes of changes in carbohydrate metabolism, and these changes can be multidirectional [35]. For example, in susceptible varieties of tomato, salt stress increased the total content of sucrose and other soluble sugars, while in moderately susceptible and tolerant varieties, it did not cause the accumulation of soluble carbohydrates [36]. 

The taxa differed toward changes in the total free amino acid content due to abiotic stresses. Whereas in *A. nipponicum* cv. Red Beauty, salinity, full sunlight, and a combination of both stresses reduced the content of amino acids, in *D. erythrosora,* the same treatments stimulated the biosynthesis of these low molecular weight organic compounds. It can be postulated that in *D. erythrosora,* total free amino acids play a role in the stress response such as proline, the accumulation of which increases under the influence of salinity in many plant species, including the water fern *Azolla filiculoides* [37]. The amino acid proline, as an osmoprotectant, reduces the water potential, preventing its outflow from the cell in response to osmotic stress caused by salinity or drought [38]. In *A. nipponicum* cv. Red Beauty, reduced total free amino acid content as a result of stress may indicate the sensitivity of this taxon to the applied stressors. A marked decrease in total essential amino acids, non-essential amino acids, and free amino acids was also shown in four edible species of ferns grown in conditions of intense sunlight [39].

### 3.3. Antioxidant Responses

On the whole, *D. erythrosora* exhibited higher total polyphenol and flavonoid content than *A. nipponicum* cv. Red Beauty. Similarly, Xia et al. [40] reported that extracts from *D. erythrosora* contained more total flavonoids than those of *A. nipponicum*. In this experiment, *D. erythrosora* plants exposed to stress were characterized by the increased production of total flavonoids, especially when exposed to full sunlight. The increase in total flavonoids in response to applied stresses appears to be an acclimation response of *D. erythrosora* to stress. Similarly, Wang et al. [7] reported increased total phenol and flavonoid content in leaves of the ferns *Matteuccia struthiopteris*, *Athyrium multidentatum*, and *Osmunda cinnamomea* var. *asiatica* exposed to the high light level. Flavanols protect plants against harmful UV-B radiation, and thus plants exposed to excess light contain much higher amounts of them than those shaded [41]. Generally, the plant limits the synthesis of secondary metabolites under favorable growth conditions due to the high energy expenditure associated with these processes. When plants are under stress, the secondary metabolism increases to balance the metabolic system [42]. Ecological factors such as light, habitat, altitude, and latitude, as well as stress factors, can activate the secondary metabolism pathways in ferns and increase the production of antioxidant compounds [7]. The confirmation of this hypothesis is the fact that extracts from the leaves of *D. erythrosora* plants grown in full sunlight had increased antioxidant activity measured by DPPH, FRAP, and reducing power tests. 

Many studies have shown that the ability to eliminate free radicals strongly correlates with polyphenol and flavonoid content [43,44]. In these studies, positive linear correlations were found in *A. nipponicum* cv. Red Beauty between total polyphenol content and activity against DPPH and between flavonoid content and activity against ABTS. In *D. erythrosora*, no correlation was found between the level of flavonoids and the values of antioxidant tests, which may be influenced by the diverse profile of phenolic compounds found in ferns. In both taxa, it was shown that the content of total free amino acids is positively correlated with the antioxidant activity measured by reducing power. Moreover, *A. nipponicum* cv. Red Beauty showed positive linear correlations between total free amino acids and the content of total polyphenols, ABTS, and DPPH. Amino acids are lesser-known antioxidants with the ability to defend against free radicals [45]. However, their role in the antioxidant system of plants is still being explored.

Fern extracts are a very rich and promising source of polyphenolic compounds with potentially beneficial medical effects [4,7,21]. Natural polyphenols and flavonoids are particularly valuable in fern tissues with documented therapeutic antioxidant, anti-inflammatory, antimicrobial, and anticancer effects [46]. The development of technologies for cultivating ferns rich in bioactive compounds and finding ways to stimulate the biosynthesis of their secondary metabolites may help in the broader use of these plants and contribute to the protection of species occurring in the natural state. It is also essential to determine the level of stress at which the yield of plants does not decrease drastically, and simultaneously, the biosynthesis of valuable metabolites increases significantly.

## 4. Materials and Methods

### 4.1. Plant Material, Treatments, and Growth Characteristics

The study materials were in vitro propagated cuttings of *Athyrium nipponicum* cv. Red Beauty and *Dryopteris erythrosora*. They were planted on 17 March 2021 into pots (0.5 dm^3^) filled with peat substrate (pH 6.0) supplemented with fertilizer 14% N, 16% P, and 18% K (0.5 g dm^−3^) and acclimatized for eight weeks in a greenhouse (22/18 °C). Well-rooted plants were transferred into larger pots (1.7 dm^3^) filled with peat substrate (pH 6.0) and fertilizer 14% N, 16% P, and 18% K (1 g dm^−3^) and placed in a plastic tunnel located near the West Pomeranian University of Technology in Szczecin (53°25′ N, 14°32′ E; 25 m above mean sea level), covered with a double layer of inflatable plastic with anti-condensation properties. The temperature in the tunnel was regulated by automatically opening the top roof ventilation when it reached 18 °C inside. Half of the plants were placed under shade screens composed of fabric with aluminum stripes, and the other half were grown without shading. All plants were arranged at a density of nine pots per square meter. The mean photosynthetic photon flux density on cloudless days at midday between 1 May and 20 June 2021 reached a mean of 103.7–273.7 μmol m^−2^ s^−1^ under shade screens and 551.9–890.1 μmol m^−2^ s^−1^ without the screens. Since 17 May 2021, half of the shaded and non-shaded (excess light) plants of both taxa were watered every seven days with 100 mmol dm^−3^ NaCl. The stress was applied four times, with 100 mL solution per pot per treatment. NaCl concentration was established based on previous studies [8]. The control was watered with tap water of pH 6.5 and electrolytic conductivity of 0.66 mS cm^−1^. Each treatment for each taxon (shade (control), full sunlight, shade + salinity, and full sunlight + salinity) included a total of 27 plants, i.e., three replicates of nine plants each. The number of sunny hours per day was: 8.2 (May), 10.2 (June), and 8.2 (July). Accumulated sunshine hours were as follows: 221.7 (May), 296 (June), and 253.6 (July).

Two weeks after the last NaCl treatment (20 June 2021), 20 plants from each treatment were assessed for the height and length of the longest frond. Then, the above-ground parts were cut and dried at 75 °C in the shade [17], and their dry weight (DW) was determined. The two most developed leaves from 10 plants from each treatment were selected for analysis.

### 4.2. Determination of Ionic Status

To determine the content of sodium, potassium, and calcium, dried and ground leaf samples were mineralized in concentrated sulfuric acid. The ion content was determined by flame photometry on a flame photometer AFP-100 (Biotech Engineering Management, Nicosia, Cyprus), as described by Ostrowska et al. [47]. Each element was determined in three replicates. The potassium/sodium (K^+^/Na^+^) and calcium/potassium (Ca^2+^/K^+^) ratios were calculated based on the data obtained.

### 4.3. Determination of Photosynthetic Pigment Content

Dried fern leaves were ground in a laboratory mill, and 0.5 g samples were placed into Falcon tubes and mixed with H_2_O/acetone mixture (2:8 *v*/*v*). The mixtures were extracted for five minutes in an ultrasonic cleaner. The samples were then centrifuged at 6000 rpm for 10 min. The content of photosynthetic pigments (chlorophyll a, b, and total chlorophyll) was determined spectrophotometrically according to the modified method of Grzeszczuk et al. [48]. In short, 300 µL of the supernatant was pipetted into a single well of a 96-well TC plate. The microplate was put into a microplate reader, and the absorbance was read at 646, 652, and 663 nm. Each sample was read in three repetitions. The content of chlorophyll a and b and total chlorophyll was expressed in mg g^−1^ DW.

### 4.4. Preparation of Extracts 

Extracts of fern leaves were obtained by solvent extraction, as described by Grzeszczuk et al. [48]. The solvent was a mixture of methanol (Merck, Darmstadt, Germany) with distilled H_2_O (7:3, *v*/*v*). Dried leaves were ground in a laboratory mill, then 0.5 g samples were transferred into 50 mL Falcon tubes (Bionovo, Legnica, Poland) and mixed with 40 mL of the mixture. The solutions were then extracted for 30 min in an ultrasonic cleaner (Elmasonic S30H, Elma Schmidbauer GmbH, Singen, Germany) and centrifuged for 5 min at 5000 rpm (Centrifuge 5418 Eppendorf, Warsaw, Poland). The extracts were filtered through 0.22 µm nylon membrane filters (Merck, Darmstadt, Germany). Three repetitions of each extract were prepared and stored in a freezer at −20 °C.

### 4.5. Determination of Total Reducing Sugars 

Total reducing sugars were determined as described by Łopusiewicz et al. [49]. Briefly, 1 mL of the supernatant was combined with 1 mL of 0.05 M acetic buffer (pH 4.8) and 3 mL of DNS reagent. The mixture was shaken vigorously and then incubated for 5 min in hot water (~96 °C). The tubes were cooled to room temperature, and the absorbance was read at 540 nm.

### 4.6. Content of Total Free Amino Acids

The total content of free amino acids was analyzed spectrophotometrically according to Łopusiewicz et al. [50]. To this end, 1 mL of the supernatant was combined with 2 mL of ninhydrin-Cd reagent in test tubes. The samples were vortexed, heated at 84 °C (5 min), and then cooled on ice. The supernatant was transferred onto a plate, and the absorbance was measured at 507 nm. The results were calculated as mg of glycine (Gly) equivalent per 1 g sample (DW).

### 4.7. Total Polyphenols and Total Flavonoids

Total polyphenols in methanolic fern extracts were determined spectrophotometrically with a microplate reader (Synergy LX, Bio Tek, Vermont, VT, USA) using the Folin–Ciocalteu method described by Tong et al. [51] with modifications. The method consisted of mixing 20 µL of the supernatant with 150 µL of distilled water and 100 µL of Folin–Ciocalteu reagent. The solution was supplemented with 80 µL of saturated Na_2_CO_3_ (after 5 min). The mixture was then incubated in the dark at 40 °C for 30 min. The absorbance was measured at 765 nm, and the concentration of the polyphenols was expressed as mg gallic acid equivalents per g of DW (mg GAE g^−1^ DW).

The total content of flavonoids was measured according to the procedure proposed by Tong et al. [42], with modifications. To this end, 25 µL of the extract was combined with 100 µL of distilled H_2_O and 7.5 µL of 5% NaNO_2_. After 5 min, the mixture was supplemented with 7.5 µL of 10% AlCl_3_ solution. The mixture was kept for 6 min at room temperature before adding 25 µL of 1 M NaOH. Then, it was diluted with 135 µL of distilled water, and the absorbance was measured at 510 nm. The results were expressed as mg quercetin equivalent (QE) per g DW (mg QE g^−1^ DW).

### 4.8. 1,1-Diphenyl-2-Picryl-Hydrazyl (DPPH) and 2,2′-Azobis(3-Ethylbenzothiazoline-6-Sulfonate) (ABTS) Assays

The DPPH and ABTS assays were performed as described by Łopusiewicz et al. [50]. First, 0.5 mL of methanolic DPPH solution (0.01 mM) was mixed with 0.5 mL of the extract in a 1:1 ratio. The samples were incubated at 22 ± 2 °C for 30 min in darkness. Following the incubation, the absorbance was read at 517 nm. The total antioxidant potential of the analyzed extracts was determined spectrophotometrically based on the ABTS assay. Before the experiment, the ABTS radical solution was diluted with 96% ethanol to obtain an absorbance of 0.700 ± 0.02 measured at 734 nm. In our experiment, we combined 1.5 mL of ABTS solution with 25 µL of the extract in 1.5 mL Eppendorf tubes, and then the tubes were incubated at room temperature for 10 min in the dark. After incubation, the absorbance was read at 734 nm.

### 4.9. Ferric Reducing Antioxidant Power (FRAP)

The FRAP reagent was prepared by mixing 2.5 mL of TPTZ in 40 mM HCl, 2.5 mL of FeCl_3_, and 25 mL of acetate buffer at pH 3.6 (300 mM). Then, 20 µL of the supernatant in three repetitions was transferred into a 96-well TC plate by adding 280 µL of the FRAP reagent, and the plate was gently shaken for 10 s. The absorbance was read at 595 nm. The results were expressed as ascorbic acid equivalents (AAE) per g of sample (DW) using the standard curve for ascorbic acid.

### 4.10. Reducing Power

The reducing power of the methanolic extracts was measured according to Łopusiewicz et al. [50], with modifications. Briefly, 100 µL of the extract was added to 1.5 mL Eppendorf tubes and supplemented with 250 µL of phosphate buffer (0.2 M, pH 6.6) and 250 µL of 1% potassium hexacyanoferrate. The samples were incubated at 50 °C for 20 min before adding 250 µL of trichloroacetic acid solution. They were then centrifuged at 3000 rpm for 10 min. In the next step, 500 µL of the solution was transferred into Eppendorf tubes (1.5 mL), and 0.5 mL of H_2_O and 0.1 mL of iron chloride (1%) were added. Finally, the absorbance was measured at 700 nm.

### 4.11. Statistical Analysis

The obtained data with normal distribution were analyzed statistically using the analysis of variance (one-way ANOVA) for one-factor experiments using the Statistica 13.1 Statsoft package (Statsoft, Krakow, Poland). After checking the good fit of the model, the post hoc comparison was completed using Tukey’s HSD post hoc test at *p* ≤ 0.05. To assess the existence of correlations between the content of metabolites and antioxidant activity, the values of Pearson’s linear correlation coefficients were determined, and their significance was verified.

## 5. Conclusions

Considering the unfavorable climate change, understanding the physiological processes responsible for the adaptive capabilities of individual plant species from different systematic groups, including ferns, is crucial. This study demonstrated that plants exposed to salinity and salinity combined with excess light produced lower bio-mass and had an unfavorable proportion of Na^+^, K^+^, and Ca^2+^ ions. In both genotypes, excess light reduced the content of chlorophyll a, chlorophyll b, and total chlorophyll. The salinity did not affect the pigment level in *D. erythrosora* and *A. nipponicum* cv. Red Beauty and, surprisingly, even stimulated their biosynthesis. A greater adaptive potential to salinity and light stress was found in *D. erythrosora* plants than in *A. nipponicum* cv. Red Beauty. The response of *D. erythrosora* to abiotic stress was strongly associated with changes in reducing the sugar content, total free amino acids, and total flavonoid content. These biochemical parameters may consequently be indicators of the sensitivity/tolerance of *D. erythrosora* to salinity and light stress. The cultivation of *D. erythrosora* under abiotic stresses contributed to the increased bioaccumulation of flavonoids in leaf tissues, which may be a new strategy for obtaining these valuable secondary metabolites in this species.

## Figures and Tables

**Figure 1 ijms-24-00296-f001:**
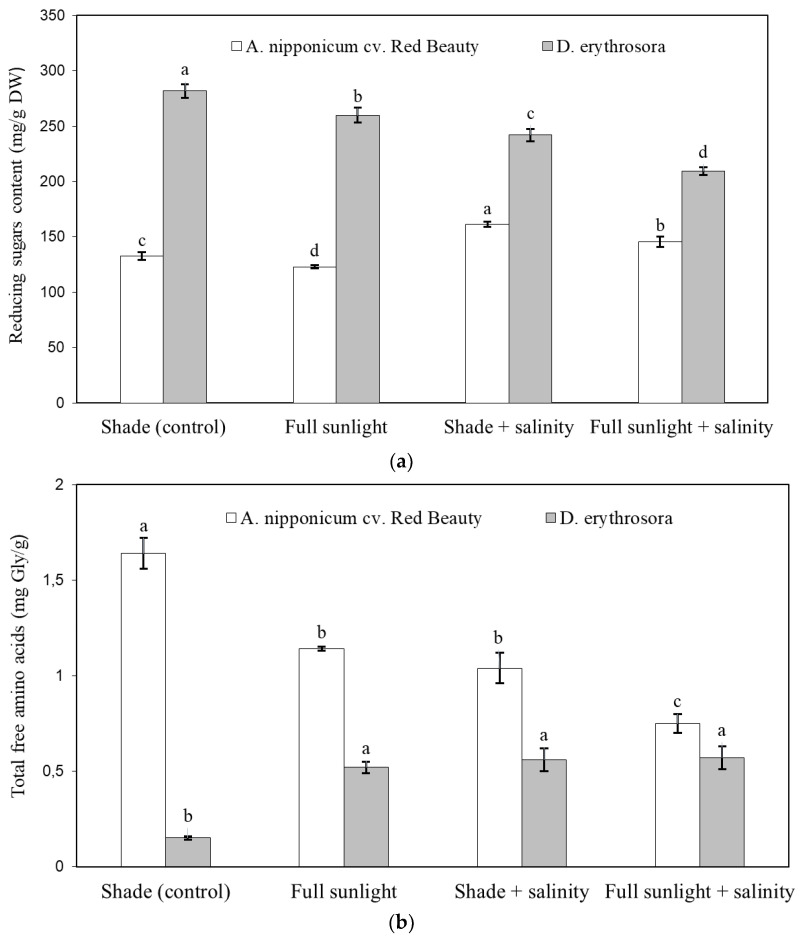
Effect of abiotic stresses on reducing sugars (**a**) and total free amino acid content (**b**) of two hardy ferns. Vertical error bars indicate the standard deviation of the mean. Different letters indicate a significant difference between treatments within given taxa at the *p* ≤ 0.05 (ANOVA, Tukey’s HSD post hoc test).

**Figure 2 ijms-24-00296-f002:**
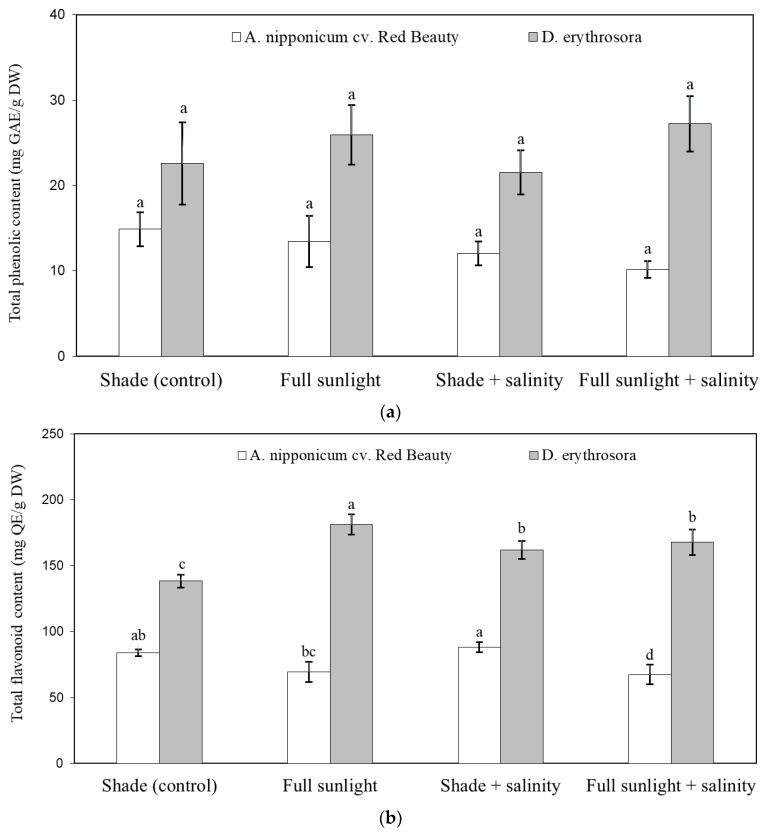
Effect of abiotic stresses on total phenolic content (**a**) and total flavonoid content (**b**) of two hardy ferns. Vertical error bars indicate the standard deviation of the mean. Different letters indicate a significant difference between treatments within given taxa at *p* ≤ 0.05 (ANOVA, Tukey’s HSD post hoc test).

**Table 1 ijms-24-00296-t001:** Effect of abiotic stresses on growth characteristics of two hardy ferns. Within a column means followed by the same lowercase letter are not significantly different at *p* ≤ 0.05 (ANOVA, Tukey’s HSD post hoc test).

Treatments	Plant Height (cm)	Frond Length (cm)	Fresh Weight (g)
*A. nipponicum* cv. Red Beauty			
Shade (control)	25.3 ± 1.07 ^a^	25.3 ± 1.06 ^ab^	8.90 ± 0.46 ^a^
Full sunlight	25.3 ± 0.96 ^a^	29.6 ± 3.92 ^a^	9.55 ± 0.42 ^a^
Shade + salinity	20.5 ± 1.50 ^b^	20.7 ± 1.47 ^c^	7.45 ± 0.18 ^b^
Full sunlight + salinity	18.8 ± 1.35 ^b^	24.1 ± 1.59 ^b^	7.47 ± 1.03 ^b^
*D. erythrosora*			
Shade (control)	24.5 ± 0.50 ^b^	37.5 ± 1.81 ^a^	14.3 ± 0.75 ^b^
Full sunlight	29.3 ± 1.53 ^a^	36.9 ± 1.41 ^a^	19.5 ± 1.16 ^a^
Shade + salinity	15.3 ± 1.53 ^c^	27.1 ± 1.90 ^b^	8.34 ± 1.30 ^c^
Full sunlight + salinity	21.7 ± 2.52 ^b^	33.7 ± 1.02 ^a^	9.75 ± 0.99 ^c^

**Table 2 ijms-24-00296-t002:** Effect of abiotic stresses excess on K^+^/Na^+^ and Ca^2+^/Na^+^ ratios of two hardy ferns. Within a column means followed by the same lowercase letter are not significantly different at *p* ≤ 0.05 (ANOVA, Tukey’s HSD post hoc test).

Treatments	K^+^/Na^+^ Ratio	Ca^2+/^Na^+^ Ratio
*A. nipponicum* cv. Red Beauty		
Shade (control)	9.00 ± 0.07 ^a^	2.97 ± 0.64 ^a^
Full sunlight	6.23 ± 0.92 ^b^	2.47 ± 0.47 ^a^
Shade + salinity	1.07 ± 0.06 ^c^	0.60 ± 0.01 ^b^
Full sunlight + salinity	1.93 ± 0.15 ^c^	0.60 ± 0.17 ^b^
*D. erythrosora*		
Shade (control)	13.47 ± 2.36 ^a^	2.97 ± 0.64 ^a^
Full sunlight	11.57 ± 1.27 ^a^	2.47 ± 0.47 ^a^
Shade + salinity	2.80 ± 0.20 ^b^	0.60 ± 0.01 ^b^
Full sunlight + salinity	2.77 ± 0.06 ^b^	0.60 ± 0.17 ^b^

**Table 3 ijms-24-00296-t003:** Effect of abiotic stresses on chlorophyll pigment content of two hardy ferns. Within a column means followed by the same lowercase letter are not significantly different at *p* ≤ 0.05 (ANOVA, Tukey’s HSD post hoc test).

Treatments	Chlorophyll a mg/g DW	Chlorophyll b mg/g DW	Total Chlorophyll mg/g DW	Chlorophyll a/b Ratio
*A. nipponicum* cv. Red Beauty				
Shade (control)	1.33 ± 0.05 ^b^	0.71 ± 0.03 ^b^	2.35 ± 0.09 ^b^	1.86 ^a^ ± 0.01 ^a^
Full sunlight	0.96 ± 0.03 ^c^	0.55 ± 0.02 ^c^	1.74 ± 0.06 ^c^	1.73 ^c^ ± 0.01 ^c^
Shade + salinity	1.52 ± 0.02 ^a^	0.86 ± 0.02 ^a^	2.75 ± 0.04 ^a^	1.78 ^b^ ± 0.01 ^b^
Full sunlight + salinity	0.80 ± 0.01 ^d^	0.47 ± 0.01 ^d^	1.46 ± 0.01 ^d^	1.71 ^c^ ± 0.03 ^c^
*D. erythrosora*				
Shade (control)	0.38 ± 0.03 ^a^	0.19 ± 0.02 ^ab^	0.63 ± 0.06 ^a^	2.01 ± 0.02 ^b^
Full sunlight	0.23 ± 0.01 ^b^	0.11 ± 0.01 ^c^	0.38 ± 0.02 ^b^	2.16 ± 0.03 ^a^
Shade + salinity	0.35 ± 0.01 ^a^	0.17 ± 0.00 ^b^	0.58 ± 0.02 ^a^	2.06 ± 0.02 ^b^
Full sunlight + salinity	0.39 ± 0.01 ^a^	0.20 ± 0.01 ^a^	0.66 ± 0.03 ^ab^	1.93 ± 0.02 ^c^

**Table 4 ijms-24-00296-t004:** Effect of abiotic stresses on antioxidant activity (as measured by ABTS, DPPH, FRAP, and reducing power methods) of two hardy ferns. Within a column means followed by the same lowercase letter are not significantly different at *p* ≤ 0.05 (ANOVA, Tukey’s HSD post hoc test).

Treatments	ABTS (%)	DPPH (%)	FRAP (mg AAE/g DW)	Reducing Power
*A. nipponicum* cv. Red Beauty				
Shade (control)	80.77 ± 0.58 ^a^	28.44 ± 1.56 ^a^	12.60 ± 0.29 ^a^	1.55 ± 0.02 ^a^
Full sunlight	60.46 ± 0.62 ^c^	20.16 ± 0.16 ^b^	9.75 ± 0.57 ^b^	1.11 ± 0.01 ^c^
Shade + salinity	76.05 ± 1.34 ^b^	17.19 ± 1.88 ^b^	9.85 ± 0.24 ^b^	1.23 ± 0.02 ^b^
Full sunlight + salinity	56.41 ± 1.31 ^d^	17.97 ± 1.41 ^b^	10.50 ± 0.50 ^b^	1.17 ± 0.04 ^bc^
*D. erythrosora*				
Shade (control)	98.62 ± 0.09 ^ab^	77.66 ± 0.47 ^b^	31.20 ± 1.77 ^b^	2.47 ± 0.03 ^b^
Full sunlight	98.15 ± 0.39 ^b^	90.16 ± 0.16 ^a^	34.91 ± 1.16 ^a^	2.55 ± 0.02 ^a^
Shade + salinity	98.97 ± 0.27 ^a^	66.72 ± 0.94 ^c^	22.43 ± 1.33 ^d^	2.50 ± 0.02 ^ab^
Full sunlight + salinity	98.92 ± 0.09 ^a^	76.88 ± 0.31 ^b^	26.09 ± 0.36 ^c^	2.53 ± 0.04 ^ab^

## Data Availability

Not applicable.

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
