# Peer review of "Changes in Growth, Ionic Status, Metabolites Content and Antioxidant Activity of Two Ferns Exposed to Shade, Full Sunlight, and Salinity"

_ijms, 2022, doi:10.3390/ijms24010296_

Round 1

Reviewer 1 Report

Pietrak and coworkers present here a study dealing with the response of two ferns species to two kind of abiotic stresses (alone and joint) linked to climate change. The authors state that one of the stresses is "excess light" (which should be 'light excess'), despite in the M&M section, they state that half of their plants were under "shade screen made of fabric with aluminum stripes" (lines 269-270).  This methodology does exactly the opposite, it reduces the available light to plants, so should be "light reduction". In addition, they state the mean PPF on cloudless days at midday between part of the experiment, but without indication of the total hours of light, how many cloudy days they get, or the total photon flux received at each group. In addition, these plants under shadow will move leaves to capture light on the edges where the shade screen is not removing light. 

Ahead of this main problem with assay design, the work also lack a bit of clarity, specially when the M&M section is at the end of the manuscript. To describe the results authors should describe really briefly what has been done at each point presented at the Results and Discussion section, and remove repetitive values that are shown also in the tables.  To this point, I would suggest authors to divide the section Results and Discussion into subheadings ie.: 2.1. Effect of abiotic stresses on fern growth parameters (to cover all the results shown in Table 1); 2.2. Effect of abiotic stresses on fern ion balance, carbohydrates and amino acid metabolism and chlorophyl content; 2.3. Effect of abiotic stresses on antioxidant compounds and antioxidant properties of ferns

This structure will clarify a bit the description of results and their impact on general knowledge. Making it easier the readiness of the manuscript, and the link with one of the conclusions drawn (strategy for production of secondary metabolism products). 

Author Response

Thank you very much for your time spent on a careful and detailed revision of our manuscript. We are grateful for numerous comments and remarks that made us reconsider many fundamental issues. Invaluable content and style related corrections allowed us to avoid multiple mistakes. In general, the review let us considerably improve our manuscript. Below you will find our answers to all the remarks. We hope our explanations are comprehensive and will dispel any possible doubts.

Pietrak and coworkers present here a study dealing with the response of two ferns species to two kind of abiotic stresses (alone and joint) linked to climate change. The authors state that one of the stresses is "excess light" (which should be 'light excess'), despite in the M&M section, they state that half of their plants were under "shade screen made of fabric with aluminum stripes" (lines 269-270).  This methodology does exactly the opposite, it reduces the available light to plants, so should be "light reduction". In addition, they state the mean PPF on cloudless days at midday between part of the experiment, but without indication of the total hours of light, how many cloudy days they get, or the total photon flux received at each group. In addition, these plants under shadow will move leaves to capture light on the edges where the shade screen is not removing light. 

Answer: We fully agree with the comments related to the determination of lighting conditions. We have adopted the following 4 treatments: Shade (control), Full sunlight, Shade + salinity and Full sunlight + salinity. We have added more information on the number of sunny hours per day and accumulated sunshine hours during plant cultivation. In our study the plants were shaded from above and from the side.

Ahead of this main problem with assay design, the work also lack a bit of clarity, specially when the M&M section is at the end of the manuscript. To describe the results authors should describe really briefly what has been done at each point presented at the Results and Discussion section, and remove repetitive values that are shown also in the tables.  To this point, I would suggest authors to divide the section Results and Discussion into subheadings ie.: 2.1. Effect of abiotic stresses on fern growth parameters (to cover all the results shown in Table 1); 2.2. Effect of abiotic stresses on fern ion balance, carbohydrates and amino acid metabolism and chlorophyl content; 2.3. Effect of abiotic stresses on antioxidant compounds and antioxidant properties of ferns.This structure will clarify a bit the description of results and their impact on general knowledge. Making it easier the readiness of the manuscript, and the link with one of the conclusions drawn (strategy for production of secondary metabolism products). 

Answer: We fully agree with the comments related to the section Results and Discussion.  These two chapters are presented separately. We used the proposed subdivisions. The text was revised.

Extensive editing of English language and style required

Answer: The professional proofreaders, who are English native speakers, have tried to keep the meaning of the text as clear and communicative as possible.

Reviewer 2 Report

Thank you for interesting job.

Author Response

Thank you for your support of our article

Reviewer 3 Report

The authors did not distinguish between the presentation of the results and their discussion. This is, hard to understand, even if it is permitted by the journal’s editorial guidelines (https://www.mdpi.com/journal/ijms/instructions). However, this presentation and discussion of the data together makes it very complex to read the work and reduces importantly its scientific value. Thus, in my opinion, a rewriting of these sections, with a clear separation of the Results from Discussion section is needed for further consideration of the paper publication.

There are many examples in the paper where this confusion between results and discussion is particularly evident: in lines, 161-163 Environmental stresses are a major cause of disturbances in carbohydrates metabolism [31]. The content of reducing sugars was varied greatly among fern taxa depending on treatment (Figure 1a). Therefore, first, a bibliographic citation is reported, and then the data of this work. Even when the method (which is incorrect in my opinion) of reporting results and discussion in the same section is used, the sequence must be: first the results and then their eventual discussion. While reporting first the discussion and then the results sounds awkward and misleading.

Lines 171-173 read: “In D. erythrosora the content of reducing sugars dropped for all treated vs. non-treated plants, with the greatest reduction (by  25.6%) noted for plants exposed to multi-stress. The taxa differed strongly in reducing sugars content as a result of stress, which may indicate genotype-specificity” But D. erythrosorav still has particularly high values in reducing sugars and that is the true genotypic specificity of the taxa, not so much the decrease under stress conditions.

Also in lines 175-181, discussions are reported first (data from other authors) and then the results of one's own work. It seems to be just the style of the authors to reverse these sections, I totally disagree with this style of writing.

Lines 178-182 highlight the differences between the taxa regarding the free aminoacids (AA) content in response to stress conditions. The authors  how inverse trends between AA and response to stress, but there is no attempt to explain this behavior, which in any case should be placed in the discussion section.

The work, although interesting in the results, is very limited in terms of explanation of the results and discussion. In conclusion, it does not seem to make any contribution to the understanding of the phenomena in progress, rather limiting itself to the simple observation of the same.

Why didn't the data from the different analyzes correlated? For instance:

-       it seems evident that minor contents in amino acids are only correlated with higher contents in polyphenols.

-       the best antioxidant activity of Dryopteris erythrosora, may be correlated with the higher content of polyphenols of these taxa

The conclusions are also simply observational, with no effort to interpret the results.

To me, the paper should be considered for publication after major revisions.

Author Response

Thank you very much for your time spent on a careful and detailed revision of our manuscript. We are grateful for numerous comments and remarks that made us reconsider many fundamental issues. Invaluable content and style related corrections allowed us to avoid multiple mistakes. In general, the review let us considerably improve our manuscript. Below you will find our answers to all the remarks. We hope our explanations are comprehensive and will dispel any possible doubts.

The authors did not distinguish between the presentation of the results and their discussion. This is, hard to understand, even if it is permitted by the journal’s editorial guidelines (https://www.mdpi.com/journal/ijms/instructions). However, this presentation and discussion of the data together makes it very complex to read the work and reduces importantly its scientific value. Thus, in my opinion, a rewriting of these sections, with a clear separation of the Results from Discussion section is needed for further consideration of the paper publication.

There are many examples in the paper where this confusion between results and discussion is particularly evident: in lines, 161-163 Environmental stresses are a major cause of disturbances in carbohydrates metabolism [31]. The content of reducing sugars was varied greatly among fern taxa depending on treatment (Figure 1a). Therefore, first, a bibliographic citation is reported, and then the data of this work. Even when the method (which is incorrect in my opinion) of reporting results and discussion in the same section is used, the sequence must be: first the results and then their eventual discussion. While reporting first the discussion and then the results sounds awkward and misleading.

Answer: We fully agree with the comments related to the section Results and Discussion.  These two chapters are presented separately. We used the proposed subdivisions. The text was revised.

Lines 171-173 read: “In D. erythrosora the content of reducing sugars dropped for all treated vs. non-treated plants, with the greatest reduction (by  25.6%) noted for plants exposed to multi-stress. The taxa differed strongly in reducing sugars content as a result of stress, which may indicate genotype-specificity” But D. erythrosorav still has particularly high values in reducing sugars and that is the true genotypic specificity of the taxa, not so much the decrease under stress conditions.

Also in lines 175-181, discussions are reported first (data from other authors) and then the results of one's own work. It seems to be just the style of the authors to reverse these sections, I totally disagree with this style of writing.

Answer: We fully agree with the comments related to the section Results and Discussion.  The text was revised.

Lines 178-182 highlight the differences between the taxa regarding the free aminoacids (AA) content in response to stress conditions. The authors  how inverse trends between AA and response to stress, but there is no attempt to explain this behavior, which in any case should be placed in the discussion section.

Answer: We fully agree with the comments. The text was revised.

The work, although interesting in the results, is very limited in terms of explanation of the results and discussion. In conclusion, it does not seem to make any contribution to the understanding of the phenomena in progress, rather limiting itself to the simple observation of the same.

Answer: The discussion chapter has been extensively revised and supplemented. We have tried to clarify the observed changes in plants. In the supplement we have added photos of the plants.

Why didn't the data from the different analyzes correlated? For instance:

-       it seems evident that minor contents in amino acids are only correlated with higher contents in polyphenols.

-       the best antioxidant activity of Dryopteris erythrosora, may be correlated with the higher content of polyphenols of these taxa

Answer: Thank you for your valuable advice. We did a correlation analysis of traits related to antioxidant potential. In the paper we described significant correlations. We have included the results in the supplement.

The conclusions are also simply observational, with no effort to interpret the results.

Answer: The conclusion of the results has been completed

Round 2

Reviewer 1 Report

The new version of the manuscript has been improved. However, in some parts, still deserves some small English polishing, but this can be omitted as the text is completely understandable, it's just a question of perfection from my point of view. 

Good work. 

Reviewer 3 Report

I thank the authors for the very detailed and challenging review. In my opinion, the work can be published in this form, even if I remain of the opinion that it remains a little speculative and very observational work. Limited contribution to the knowledge of the physiological mechanisms involved.